# Barriers to Compliance with National Guidelines Among Children Hospitalized with Community-Acquired Pneumonia in Vietnam and the Implications

**DOI:** 10.3390/antibiotics14070709

**Published:** 2025-07-15

**Authors:** Thuy Thi Phuong Nguyen, Huong Thi Thu Vu, Anh Minh Hoang, An Minh Ho, Israel Abebrese Sefah, Brian Godman, Johanna C. Meyer

**Affiliations:** 1Faculty of Pharmaceutical Management and Economics, Hanoi University of Pharmacy, Hanoi City 10000, Vietnam; anh.hoangminh.rg@gmail.com (A.H.M.); hominhan1604@gmail.com (A.H.M.); 2E Hospital, Hanoi City 10000, Vietnam; vtthuongbve@gmail.com; 3School of Pharmacy, University of Health and Allied Sciences, Ho PMB 31, Volta Region, Ghana; isefah@uhas.edu.gh; 4Department of Public Health Pharmacy and Management, School of Pharmacy, Sefako Makgatho Health Sciences University, Ga-Rankuwa 0208, South Africa; hannelie.meyer@smu.ac.za; 5Strathclyde Institute of Pharmacy and Biomedical Sciences, University of Strathclyde, Glasgow G4 0RE, UK; 6Antibiotic Policy Group, City St. George’s, University of London, London SW17 0RE, UK; 7South African Vaccination and Immunisation Centre, Sefako Makgatho Health Sciences University, Ga-Rankuwa 0208, South Africa

**Keywords:** adherence, antibiotics, antimicrobial stewardship, children, community-acquired pneumonia, hospitals, national guidelines, Vietnam

## Abstract

**Background**: Community-acquired pneumonia (CAP) is the leading cause of death in infants aged 1–59 months. Concurrent with this, there is a need to prescribe antibiotics wisely in Vietnam due to concerns with rising antimicrobial resistance (AMR). Consequently, an urgent need has arisen to treat patients according to agreed guidelines. The aim of this study was to investigate the current management of infants under five years old with CAP in Vietnam as well as identify possible obstacles to adhering to national guidelines. **Methods**: A mixed-method approach was used incorporating both quantitative and qualitative data analysis in a leading hospital in Vietnam, which influences others. Data from 108 pediatric patient records were collected and analyzed. Subsequently, in-depth interviews were conducted with pediatric doctors treating these patients to ascertain possible reasons for non-adherence to guidelines. **Results**: The mean age of children diagnosed with CAP was 27.94 ± 12.99 months, with 82.4% having non-severe CAP, and 41.7% of children had previously used antibiotics before hospitalization. The median length of hospital stay was 7 days. All children were prescribed antibiotics, 91.4% of children received these initially intravenously, with third-generation cephalosporins being the most (91.7%) commonly prescribed. Cefoperazone/sulbactam was the most frequently prescribed (48.2%) antibiotic. However, on 96.1% of occasions cefoperazone/sulbactam was given at higher doses than the label instructions. Overall, 73.3% of antibiotics prescribed were “Watch” antibiotics. In addition, the proportion of initial antibiotic regimens that were consistent with current national guidelines was only 4.63%. **Conclusions**: There were considerable concerns with low adherence rates to current guidelines alongside high rates of prescribing of injectable third-generation cephalosporins due to various internal and external barriers. Antimicrobial stewardship programs with updated national guidelines are urgently needed in Vietnamese hospitals to treat CAP in children as part of ongoing measures to reduce increasing AMR rates. Such activities should also help improve antibiotic use in the community following improved education of trainee ambulatory care physicians regarding appropriate management of children with CAP.

## 1. Introduction

Whilst the number of deaths in children under 5 years have been decreasing worldwide during the past 30 years, lower respiratory tract infections (LRTIs), including pneumonia, are still the most dominant causes of immediate death [1,2].

Among infants and children aged between 1 and 59 months, community-acquired pneumonia (CAP) is a leading cause of death at 15.5% of all deaths, second only to preterm birth complications globally [2,3]. This is particularly the case among infants and young children in LMICs, especially those in sub-Saharan Africa and Southern Asia [2,3,4,5]. High mortality rates need addressing as there is still considerable preventable morbidity and mortality from childhood pneumonia in LMICs including Vietnam [3,6,7,8]. Overall, childhood CAP is still a substantial burden on healthcare systems in LMICs including Vietnam [4,8,9].

Alongside this, there is growing antimicrobial resistance (AMR) globally enhanced by considerable overuse of antibiotics, especially among LMICs, increasing morbidity and mortality from infectious diseases [10,11,12,13]. Initiatives to address this public health concern include the WHO Global Action Plan to reduce AMR launched in 2015, and subsequently translated into National Action Plans (NAPs) [14,15]. This includes Vietnam, which was the first country in the WHO Western Pacific Region to develop its NAP to combat AMR [16,17]. Other WHO initiatives include encouraging the classification of antibiotics into Access, Watch and Reserve (AWaRe) groups depending on their resistance potential [18,19], followed by the publication of the WHO AWaRe guidance in 2022 [8,20,21,22]. The emphasis is on encouraging the preferential prescribing of Access antibiotics where pertinent [19,23], with the target for Access antibiotics increased to 70% of total antibiotic consumption across sectors following the United Nations General Assembly (UN GA) for AMR in 2024 [24].

Despite Vietnam being the first country in the WHO Western Pacific Region to develop its NAP to combat AMR, there are concerns with its implementation [16,17]. This includes continued overuse of antibiotics, especially those with appreciable resistance potential, exacerbating high rates of AMR [25,26,27,28,29,30,31]. Antibiotics from the WHO Watch list now account for 56% of total antibiotic use in Vietnam, highest among national hospitals ranging from 72.4% to 76.9% of total consumption [25,32]. Whilst we see high use of Watch antibiotics in other LMICs, this trend needs to be urgently addressed to attain UN GA goals for increasing Access antibiotic consumption [23,24,32].

In their study regarding the management of children in Vietnamese hospitals with CAP, Tran et al. (2024) ascertained that 95% of children were treated with antibiotics, mostly with cephalosporins [5]. There were also issues with the prolonged use of antibiotics, as long as 22 days in some children [5], with short courses, i.e., 3 to 5 days, increasingly advocated to reduce unnecessary antibiotic prescribing unless children have severe pneumonia [33,34,35,36]. Bacterial co-infections are also seen among hospitalized children with CAP in Vietnam and other LMICs, especially those with severe pneumonia [5,7,37,38], increasing antibiotic prescribing.

There have also been concerns with unnecessary and prolonged use of intravenous antibiotics among children with CAP in Vietnam [26,39], coupled with a lack of step-down care to oral antibiotics [39]. Alongside this, there are generally high rates of prescribing of antibiotics for neonates and children admitted to a range of hospitals in Vietnam (Appendix A) [40,41,42]. This includes neonates and children with pneumonia as well as severe pneumonia. There were also high levels of antibiotic use prior to hospital admission (Appendix A) [40,41,42], similar to other LMICs [43,44]. This also impacts antibiotic choices once hospitalized, resulting in high levels of empiric prescribing of antibiotics, including broad-spectrum antibiotics, among patients with CAP in Vietnam [40,41,42,45]. The WHO AWaRe prescribing guidance book suggests simplifying treatment of hospitalized patients with CAP to narrow spectrum antibiotics where possible based on culture findings to reduce AMR [8,21].

The urgent need to reduce excessive inappropriate prescribing of antibiotics among children with CAP in Vietnam is illustrated by current high rates of multi-drug resistance (MDR) to *Streptococcus pneumoniae* in these patients. Resistance rates rose from 31% to 80% in a 15-year study period among children in rural Vietnam [46]. Similarly, Dai et al. (2020) found high rates of MDR *pneumococci* in the community among infants enrolled into a pneumococcal conjugate vaccine trial [47]. Tran-Quang et al. (2023) also found *Streptococcus pneumoniae* isolates from children hospitalized in Vietnam with CAP were resistant to many antibiotics [48].

There are also concerns with increased resistance to co-amoxiclav and cefuroxime, alongside increased resistance to *Mycoplasma pneumoniae*, among patients hospitalized with pneumonia in Vietnam [49,50,51]. High levels of MDR strains have also been seen generally in patients with lower respiratory tract infections (LRTIs) admitted to hospitals in Vietnam [52]. MDR strains are a concern as these can lead to an increase in re-admission of patients with CAP in Vietnam unless patients are appropriately managed [53]. More recently, Tran et al. (2023) identified *Methicillin-resistant Staphylococcus aureus* (MRSA) strains as the second leading cause of severe CAP among children in Vietnam hospitalized with this condition [54]. Of concern as well is that MRSA isolates in their study were completely resistant to penicillin (100%) as well as resistant to many other antibiotics including clindamycin (84.4%) and erythromycin (78.1%) [54]. High levels of MRSA were also seen in the study of Vo-Pham-Minh et al. (2024) among patients hospitalized with CAP [55].

Antimicrobial stewardship programs (ASPs) have improved the use of antibiotics in hospitals across countries helping to reduce AMR [56]. However, there have been concerns that ASPs are more difficult to perform in LMICs due to personnel and financial issues, which includes Vietnam [31,57]. This though is changing, and we are seeing multiple ASPs being performed across LMICs, including Vietnam, to improve future antibiotic use [12,58,59,60,61,62,63,64]. This includes among patients with acute respiratory tract infections and CAP [62,63,64]. Compliance with agreed guidelines, including patients with CAP, is increasingly part of ASPs in LMICs to improve future antibiotic use and the quality of care [58,65,66,67,68]. It is increasingly likely that quality targets for managing hospitalized patients with infectious diseases, including CAP, will be based on percentage adherence rates to prescribing recommendations, including within the WHO AWaRe guidance book, to meet UN GA targets [8,20,21,24,69].

In their study, Nguyen et al. (2023) found there was a reduction in antibiotic use among young children hospitalized with respiratory infections in Vietnam after physician training with evidence-based management algorithms [70]. This is important with concerns regarding the knowledge and implementation of ASPs among pediatricians and others in LMICs including Vietnam [64,71,72], which have resulted in requests for AMS training and education among physicians in Vietnam [65]. Encouragingly, studies in Vietnam have shown there was greater confidence in the prescribing antibiotics among physicians in tertiary hospitals in Vietnam with their additional training, with typically positive attitudes towards AMS [73].

There have also been cultural concerns and other challenges, including behavioral challenges, that need to be addressed for the successful implementation of ASPs in Vietnam. This includes improving the management of patients with CAP with increased adherence to guidelines [74], similar to other LMICs [75,76,77,78,79]. Ensuring hospital-based pediatricians prescribe antibiotics appropriately is important as they are training the prescribers of tomorrow, including those treating neonates and children in the community [71].

We wanted to build on earlier studies in Vietnam (Appendix A) demonstrating concerns with the management of CAP among young hospitalized children with CAP, especially antibiotic use. In addition, identify possible barriers to the increased use of national guidelines to improve the future care of young children with CAP in Vietnam. Box 1 gives details of the current recommendations for managing neonates and children hospitalized with CAP in Vietnam [80].

Box 1Current guidelines for diagnosis and management of young children in Vietnam.
Amoxicillin is the first-line antibiotic of choice, with amoxicillin/clavulanic acid, cefuroxime, cefaclor, erythromycin and azithromycin listed as second-line alternativesFor the treatment of severe pneumonia in children: Penicillin A with an aminoglycoside antibiotic. If initial treatments fail, ceftriaxone or cefotaxime can be administered intravenously.Antibiotics are recommended for a minimum period of 5 days, with intravenous antibiotics reserved for children who experience pneumonia-related complications (e.g., empyema) or a poor response to oral antibiotics.Intravenous to oral step-down antibiotic is advised following adequate clinical recovery and once oral antibiotics can be tolerated


Identifying barriers to improve compliance with national guidelines is seen as particularly important, with adherence to guidelines known to improve antibiotic use and outcomes in patients with CAP in LMICs [67,68,70,81]. Consequently, it is critical to understand current prescribing patterns for CAP among infants and young children in Vietnam, alongside potential barriers to the increased use of agreed guidelines among leading pediatricians, to improve the rational use of antibiotics in this population [82]. These were the objectives of this study, with the findings used to provide future direction to all key stakeholder groups in Vietnam.

## 2. Results

Most of the 108 children in the study population were in the age group > 24 to 36 months, with more males (50.9%) than females. Notably, 41.7% of the children had a history of using antibiotics before hospitalization, with 7 (6.5%) consuming more than one kind of antibiotic before admission, and in 10 cases (9.3%) using a third-generation cephalosporin (Table 1).

Overall, 82.4% of hospitalized children had non-severe pneumonia, while 60.2% had no concurrent illnesses. 100% of the hospitalized children were prescribed antibiotics (Table 2). This included children with both non-severe and severe CAP. 74.1% of children were prescribed monotherapy, which could include a fixed-dose combination (FDC), with 25 admitted children (23.1%) prescribed two antibiotics either alone or as an FDC combined with another antibiotic. Three neonates and children were administered three antibiotics, i.e., three individual antibiotics or as an FDC plus two other antibiotics (2.8%) (Table 2). Higher rates of combinations of antibiotics were prescribed among children admitted with severe pneumonia (31.6% vs. 24.7%) (Table 2).

Over 90.0% of hospitalized children treated initially with antibiotics were prescribed regimens containing at least one third-generation cephalosporin (Table 2). Cefoperazone/sulbactam (38.0%) and cefotaxime (19.4%) were the most commonly prescribed antibiotics as monotherapy (Table 2). Cefoperazone/sulbactam was also the most common antibiotic administered among children initially prescribed multiple antibiotics (39.4% –10.2%/25.9%)—Table 2.

Typically, the antibiotics were administered intravenously (91.4%), with oral antibiotics administered initially in only 8.6% of infants and young children, which was typically a macrolide.

Children were prescribed antibiotics on a median of 6.00 (5.25–7.00) days, and typically discharged on a median of 7.00 (6.00–8.00) days following admission either because their symptoms had disappeared; alternatively, there was sufficient improvement to warrant discharge (Table 1).

Dosing regimens for initial monotherapy were typically higher than recommendations for infants and children with severe pneumonia and pneumonitis who were administered an FDC initially compared with those administered a single antibiotic (89.1%—49/55) (Table 3). Overall, 73.3% of antibiotics prescribed initially were Watch antibiotics (Table 3), and 5.7% were unclassified.

Only 2 cases of step-down therapy from IV to oral were identified. The first one was due to a previous IV-line failure; consequently, a new one could not be inserted. The second switch was because of significant patient improvement; consequently, IV administration was seen as no longer required.

Overall, only 42.2% of the antibiotics were administered at the recommended dosage, with 40.7% of infants and young children prescribed antibiotics at higher than recommended dosages (Table 3). Physicians’ initial antibiotic choices were heavily influenced by the patient’s antibiotic history before hospital admission as well as available antibiotics in the hospital, with inconsistent antibiotic availability affecting prescribing options.

92.6% of patients had culture tests undertaken (Table 4). When undertaken, 100% of samples were taken within the first day of hospitalization. Consequently, initial antibiotic treatment was empiric, not helped by often long time periods before sensitivity reports were available (Table 4).

However, only two patients were subsequently prescribed an alternative antibiotic following the sensitivity test results, limiting their influence in practice.

Overall, only 4.63% of antibiotics prescribed were consistent with current national guidelines. A number of barriers were identified impacting on the use of national guidelines (Table 5). These included concerns that prior antibiotic use is not sufficiently addressed and that current national guidelines are outdated and inappropriate for the current situation to treat infants and young children in hospital facilities.

## 3. Discussion

The children involved in this study in Vietnam were aged between 2 months to five years, representing a demographic with one of the highest mortality rates due to pneumonia as reported by the WHO [2]. Consequently, it is essential to ensure optimal management.

There was a high rate of prior antibiotic treatment (41.7%) among admitted children, similar to previous studies conducted in Vietnam (Appendix A), as well as other studies across LMICs [5,26,39,40,41,43]. This high use of antibiotics in the community may reflect greater knowledge among primary care healthcare professionals (HCPs) surrounding the management of respiratory infections. However, further research is needed before we can say anything with certainty especially as ten children were prescribed a third-generation cephalosporin before admission. In any event, it is essential that hospital physicians initially treating neonates and children admitted with CAP take a full history, including prior antibiotic use, to guide initial antibiotic use in the absence of any culture and sensitivity findings.

All children (100%) subsequently admitted to hospital were prescribed antibiotics, principally intravenously (91.7%), and typically Watch antibiotics (73.3%), similar to other studies conducted in Vietnam and other LMICs [5,26,39,41,42,43,44,83]. This situation needs urgent changing alongside current low compliance levels with CAP guidelines. The low compliance levels seen in this study were substantially lower than seen in a number of other studies, including previous studies in Vietnam, which is a concern [26,68,84]. We believe the principal reasons for the low adherence rates to current guidelines seen in this study could be due to concerns with the availability of appropriate antibiotics in hospitals in Vietnam [85] alongside widespread use of pre-hospitalization antibiotics (Appendix A) [40,41,42]. Encouragingly, guideline compliance, coupled with improved antibiotic selection in patients with CAP, including pediatric patients, have improved following ASPs [86,87,88,89,90]. All these studies providing direction to key stakeholder groups in Vietnam treating children admitted to hospital with CAP to improve future guideline compliance.

High rates of IV antibiotics initially (91.4%) are also a concern, and against current guidance (Box 1). We have seen high rates of IV antibiotic administration in other studies in Vietnam as well as other LMICs among neonates and children including those with CAP [26,39,91,92,93]. Excessive IV administration needs to be avoided where possible as this can result in pain at injection site, issues with patient compliance, phlebitis along with local and systemic infections, as well as potentially increasing hospital length of stay and costs [94,95]. Early switching to oral antibiotics where possible as part of ASPs can help address this [94,95,96,97]. Overall though, the length of antibiotic prescribing was within current guidelines with a median of 6 days with a minimum of 5 days recommended (Box 1). However, some patients received longer courses (Table 1), which contrasts with the WHO AWaRe book guidance suggesting that 3 to 5 days of antibiotic treatment among children hospitalized with CAP may be sufficient [8,20,21].

Several barriers were identified that need to be addressed to improve future adherence to current CAP guidelines among hospitals in Vietnam. These include the need to update current national guidelines, which were last updated in 2015. Since then, there has been a continual rise in AMR in Vietnam, which will impact on potential antibiotic choices [47,48,50,51,54]. The WHO AWaRe treatment guidance book is a good starting point, amended based on local AMR patterns [8,20,21]. Updated guidelines must include treatment recommendations based on prior antibiotic use in children with CAP given the high prevalence rates currently seen in Vietnam (Appendix A). The guidelines should also be readily accessible, and easy to understand, to enhance their subsequent utilization [13].

The classification system for patients admitted with CAP in Vietnam also needs to be reviewed, including clinical symptoms on admission, given concerns among some of the interviewees (Table 5). Subsequently, potential quality indicators (QIs) can be agreed among all key stakeholder groups as part of ASPs to improve the management of neonates and children hospitalized with CAP in the country [60,69]. This can be accompanied by appropriate educational initiatives as part of ASPs given successes with these initiatives among LMICs, including children with CAP [58,62,63,64,67,68,70]. Physician education can start among Universities in Vietnam and continue post-qualification as part of continuous professional development activities [58,98]. This especially with studies in Vietnam showing greater confidence in prescribing antibiotics among physicians in tertiary hospitals in Vietnam with positive attitudes towards AMS and ASPs [73].

Finally, issues of shortages of appropriate antibiotics need to be addressed going forward where this occurs [99,100]. This can start with key stakeholders in each hospital in Vietnam treating neonates and children with CAP agreeing on alternative antibiotics when there are shortages [101]. Subsequently, working with Ministry of Health personnel to address key issues and challenges surrounding shortages of key antibiotics including potentially updating forecasting systems in hospitals [101,102].

We are aware of some limitations with this study. Firstly, the data was collected retrospectively from handwritten medical records without the availability of electronic medical records. Consequently, it is possible that the clinical characteristics of neonates and children described in the medical records may not fully reflect their actual clinical condition. However, this is common for retrospective analysis of patients’ notes. Secondly, the classification of the severity of CAP in children was performed by the researcher and may not reflect the actual situation. Thirdly, we only collected data from E Hospital. However, this study built on previous CAP studies among this population in Vietnam (Appendix A). In addition, E Hospital is a Grade I central general hospital with over 1000 beds; consequently, if there are problems with the management of neonates and children in this hospital, these are likely to be replicated across Vietnam. Finally, we are aware that we only interviewed a limited number of physicians. However, this was the total number of pediatricians treating neonates and children in the hospital at the time. Despite these limitations, we believe our findings are robust offering guidance to all key stakeholder groups going forward to improve the care of this vulnerable population.

## 4. Materials and Methods

### 4.1. Research Design and Subjects

We adopted a mixed-methods approach including a retrospective data review of the management of children under 5 years of age hospitalized for CAP at E Hospital post the COVID-19 pandemic. This was followed by in-depth interviews with key physicians managing these patients in E Hospital regarding possible barriers to the use of national guidelines to manage children hospitalized with CAP in their facility.

E Hospital was chosen for this updated study as it is a Grade I central general hospital in Hanoi, Vietnam, under the Ministry of Health, established under the Prime Minister in 1967. Currently, the hospital has over 1000 beds, including a cardiovascular center, and 62 functional departments, and receives health insurance patients from across Vietnam. Consequently, if there are problems with the management of neonates and children in this hospital, these are likely to be replicated across Vietnam.

The Pediatrics Department in E Hospital was established in 2018 and has been in continual operation since then. According to hospital summary report in 2022, the hospital has 7 groups of antibiotics available for prescribing containing 84 individual antibiotics, with the cost of antibiotics accounting for 30.65% of the total cost of medicine annually consumed in the hospital.

The appropriateness of antibiotic prescriptions was evaluated according to the latest Vietnam national guidelines (Box 1) [80]. This included the initial antibiotic choices, route of administration, dosage, frequency, duration, and length of hospital stay. Appropriate dosages of antibiotics were assessed based on the product information contained within the Vietnam National Pharmacopoeia. There were 3 cases where information about the child’s weight was missing; consequently, the appropriate dosage assessment could not be performed.

After obtaining the quantitative results, in-depth interviews were conducted with five doctors directly involved in treating these patients in E Hospital to identify potential barriers to compliance with national guidelines. Physicians were selected through purposive sampling. To be eligible for inclusion in the study, the physicians had to be currently employed in the hospital at the time of the study, be directly involved in treating these patients and provide consent to be interviewed in depth.

### 4.2. Data Collecting and Processing Methods

A data collection form for this study was developed based on the experiences of the co-authors to collect relevant data regarding the management of children under 5 hospitalized with CAP in E Hospital. We have used this approach before across LMICs [103,104,105]. The relevant patient information was subsequently inserted into the Medical Record Information Collection Form, designed on REDCap version 13.4.2. Information retrieved from the medical record included the medical record code, the general characteristics of the child (name, age, gender, weight, diagnosis, treatment duration, comorbidities, medical history, allergy history, history of antibiotic use before hospitalization, clinical and paraclinical characteristics, clinical symptoms, severity, and microbiological test results [if any]). In addition, details of antibiotics prescribed for CAP for the child, e.g., type of initial antibiotic regimen, number of antibiotics used in the initial regimen, route of antibiotic administration, antibiotic dosage, duration of initial regimen, characteristics of changes in the initial antibiotic regimen (if any). Alongside this, details of length of hospital stay; treatment results; name of the main treating physician; and the severity of pneumonia classified according to the standards guided by the Vietnam Ministry of Health [80]. The appropriateness of the initial antibiotic regimen was assessed in terms of whether the antibiotic selected belonged to one of the regimens recommended by the Vietnam Ministry of Health [80]. The antibiotics prescribed were also broken down by the WHO AWaRe classification published in 2019 [19], with compliance assessed by the extent of actual compliance with the initial antibiotic regimen selection for treating children with CAP according to the guidelines of the Vietnamese Ministry of Health [80].

There was no sample size calculation as the entire medical records of children under 5 treated at the Pediatrics Department, E Hospital, from 1 January 2021, to 31 December 2022, were reviewed.

The inclusion criteria included the following:

Medical records of patients with a confirmed diagnosis of pneumonia (ICD code of discharge diagnosis is J12 to J18);Medical records of patients aged 2 months—5 years;Medical records of patients with indications for antibiotic use within 48 h of admission;Medical records of inpatients for 3 days or more. This date was chosen to ensure that the selected cases were truly cases of CAP and patients had undergone a sufficiently long initial treatment period to allow meaningful analysis, i.e., helping to exclude cases of uncertain diagnosis. This is because we were aware that some children are admitted to hospitals in Vietnam with symptoms resembling pneumonia; however, this may be another infectious disease that quickly resolves. If the child is discharged or stops treatment within 1–2 days, it is likely that the initial diagnosis of pneumonia was incorrect. Consequently, setting a threshold of 3 days helps to eliminate these cases making the study patient group more homogeneous and robust.

The exclusion criteria included the following:

Medical records of young children in which the diagnosis of pneumonia was not recorded within the first 48 h of admission;Medical records of patients who were subsequently transferred to another hospital;Medical records that could not be accessed.

The data was subsequently analyzed using Microsoft Excel 2003 and IBM SPSS Statistics 20. A total of 108 qualified medical records were involved in the quantitative research to determine the characteristics and appropriateness of antibiotic use in treating children under 5 with CAP in the pediatric department at E Hospital.

The interview guide was developed based on the literature alongside the considerable experience of the co-authors conducting studies of this nature to provide future guidance. We have used this method before when conducting interviews with key stakeholders to provide future direction [106,107,108].

All five pediatricians in the hospital were approached for interviews and all five agreed to participate. This represented 100% of the total number of pediatricians treating children with LRTIs in E Hospital. The pediatricians gave their informed consent verbally to be interviewed and that the interviews could be taped prior to their interview to enhance analysis.

The findings from the in-depth interviews were subsequently transcribed, imported into NVivo version 10, and analyzed by thematic analysis. Two researchers independently coded and discussed the findings to reach a consensus to ensure reliability and robustness in the findings. The analysis was guided by an inductive approach, with themes drawn from the data itself.

### 4.3. Ethical Approval

The study was approved by the Institutional Review Board of the E hospital (No. 1054/QD-BVE dated 18 April 2023) and the ethics committee of E hospital (No.159/PCT-HĐĐĐ) (Appendix A).

## 5. Conclusions

There were concerns with low adherence to current CAP guidelines for the management of hospitalized neonates and children in this hospital in Vietnam, reflecting the situation previously in other hospitals in Vietnam. Alongside this, issues with high rates of parenteral antibiotic administration with limited evidence of step-down to oral therapy. There were also concerns that current CAP guidelines published in 2015 are now out-of-date. Urgent activities are needed to improve future prescribing, and reduce AMR, in the country among this population.

Initial suggested activities include the Ministry of Health updating the national guidelines since these were last updated in 2015 and there has been considerable growth in AMR since then. Future guidelines for children hospitalized with CAP should be based on the WHO AWaRe guidance, modified for local resistance patterns, and include re-appraising the classification system given current concerns. Subsequently updated every 5 years to reflect any changes in antimicrobial resistance patterns in Vietnam. Once updated, educational activities need to be instigated among all key stakeholder groups to review and implement the updated guidelines. The Pharmacy Departments in hospitals can play a lead role in this respect through helping to undertake training of key HCPs within the hospital as well as setting warnings with prescribers regarding appropriate dosing of antibiotics in neonates and children, potentially enhanced by prescription software. Training needs to include suggested changes in antibiotics prescribed depending on prior hospital administration given the substantial use of antibiotic pretreatment in children with CAP currently seen in Vietnam. Pharmacy Department personnel can also play an active part with encouraging early switching from IV to oral antibiotics were pertinent to reduce costs and hasten earlier discharge. This can be part of appropriate ASPs undertaken in this and other hospitals in Vietnam treating children with CAP. ASPs can include instigating agreed quality indicators to improve the future management of these vulnerable children. Alongside this, Pharmacy Departments in hospitals need to actively research the nature and extent of any shortages of antibiotics critically important in managing patients with CAP given current concerns. Subsequently, seek to address any key identified challenges including concerns with forecasting and ordering of antibiotics especially with the instigation of appropriate software in hospitals in Vietnam including prescribing software.

Universities in Vietnam also need to play their part with improving the care of children hospitalized with CAP in Vietnam by critically appraising their current curricula to ensure trainee physicians leave their university training fully conversant with the AWaRe classification, appropriate management of neonates and children with CAP, and the need to follow national guidelines as part of ASP. Trainee hospital pharmacists also need to be fully conversant with similar issues, especially if they take the lead with ASP activities in hospitals including appropriate dosing of antibiotics and earlier switching to oral antibiotics. Such activities should also help improve subsequent antibiotic use in the community as the physicians trained in hospitals subsequently practice in the community.

## Figures and Tables

**Table 1 antibiotics-14-00709-t001:** Characteristics of the study population, antibiotic use and outcomes (*n = 108*).

Characteristics		*n*	*%*
Age (months)	27.94 ± 12.99		
Gender *(n = 108*)	Male (*n*,%)	55	50.9
	Female (*n*,%)	53	49.1
*Medical history—antibiotic use*			
Antibiotics use pre-admission	Yes	45	41.7
	No	55	50.9
	N/A	8	7.4
*Clinical state on admission*			
Severity	Non-severe pneumonia (*n*,%)	89	82.4
	Severe pneumonia (*n*,%)	19	17.6
Comorbidity	Yes (*n*,%)	43	39.8
	No (*n*,%)	65	60.2
*Treatment duration and effectiveness*			
Antibiotic treatment duration (days—Median)		6.00 (5.25–7.00)	
Hospitalization duration (days—Median)		7.00 (6.00–8.00)	
Outcome	Recover	63	58.3
	Reduced symptoms	45	41.7

**Table 2 antibiotics-14-00709-t002:** Characteristics of the initial antibiotic regimens and routes of administration.

Initial Antibiotic Regimen	Non-Severe Pneumonia		Severe Pneumonia		Total	
	*n*	%	*n*	%	*n*	%
** *Monotherapy* ** ** *(including combinations)* **	** *67* **	** *75.3* **	** *13* **	** *68.4* **	** *80* **	** *74.1* **
*Penicillin/β-lactamase*						
Amoxicillin/Sulbactam (IV)	5	5.6	1	5.3	6	5.6
Amoxicillin/Clavulanic (IV)	1	1.1	0	0.00	1	0.9
Ampicillin/Sulbactam (IV)	0	0.00	1	5.3	1	0.9
*Third-generation cephalosporin/β-lactamase combination*						
Cefoperazone/Sulbactam (IV)	34	38.2	7	36.8	41	38.0
*Third-generation cephalosporins*						
Cefotaxime (IV)	18	20.2	3	15.8	21	19.4
Cefotiam (IV)	3	3.4	1	5.3	4	3.7
Ceftizoxime (IV)	5	5.6	0	0.00	5	4.6
Macrolides					1	0.9
Clarithromycin (O)	1	1.1	0	0.00	1	0.9
** *Combination regimens* **	** *22* **	** *24.7* **	** *6* **	** *31.6* **	** *28* **	** *25.9* **
*Third-generation cephalosporins/inhibitor β-lactamase + Aminoglycoside*						
Cefoperazone/Sulbactam (IV) + Amikacin (IV)	5	5.6	2	10.5	7	6.5
*Third-generation cephalosporins/inhibitor β-lactamase + Macrolides*						
Cefoperazone/Sulbactam (IV) + Azithromycin (O)	1	1.1	1	5.3	2	1.9
Cefoperazone/Sulbactam (IV) + Clarithromycin (O)	1	1.1	0	0.00	1	0.9
*Third-generation cephalosporins + Aminoglycosides*						
Cefotaxims (IV) + Gentamicin (IV)	3	3.4	2	10.5	5	4.6
Cefotaxime (IV) + Amikacin (IV)	1	1.1	0	0.00	1	0.9
Ceftizoxime (IV) + Amikacin (IV)	3	3.4	0	0.00	3	2.8
Cefotiam (IV) + Amikacin (IV)	1	1.1	0	0.00	1	0.9
*Third-generation cephalosporins + Macrolides*						
Cefotaxime (IV) + Azithromycin (O)	2	2.3	0	0.00	2	1.9
Cefotiam (IV) + Azithromycin (O)	1	1.1	0	0.00	1	0.9
Cefotaxime (IV) + Clarithromycin (O)	2	2.3	0	0.00	2	1.9
*Third-generation cephalosporins/inhibitor β-lactamase + Aminoglycosides + Macrolides*						
Cefoperazone/Sulbactam (IV) + Amikacin (IV) + Clarithromycin (O)	0	0.00	1	5.3	1	0.9
*Third-generation cephalosporins + Aminoglycosides + Macrolides*						
Ceftizoxime (IV) + Amikacin (IV) + Clarithromycin(O)	1	1.1	0	0.00	1	0.9
Cefotiam (IV) + Gentamicin (IV) + Azithromycin(O)	1	1.1	0	0.00	1	0.9
** *Total* **	89	100.00	19	100.00	108	100.00

NB: O = oral; IV = intravenous injection.

**Table 3 antibiotics-14-00709-t003:** Antibiotic use characteristics—initial management (*n = 105*).

*Initial Antibiotics*	*Classification **	*Route of Drug Administration*	*Daily Dosage*					
			Lower		Appropriate		Higher	
			*n*	%	*n*	%	*n*	%
Amikacin	A	IV	1	7.1	12	85.7	1	7.1
Amoxicillin/Sulbactam	N/A	IV	1	16.7	5	83.3	0	0.0
Amoxicillin/Clavulanic	A	IV	0	0.0	1	100.0	0	0.0
Ampicillin/Sulbactam	A	IV	0	0.0	1	100.0	0	0.0
Azithromycin	W	O	3	60.0	0	0.0	2	40.0
Cefoperazone/Sulbactam	W	IV	0	0.0	2	3.9	49	96.1
Cefotaxime	W	IV	12	38.7	19	61.0	0	0.0
Cefotiam	W	IV	1	16.7	5	83.33	0	0.0
Ceftizoxime	W	IV	4	44.4	5	55.6	0	0.0
Clarithromycin	W	O	0	0.0	3	60.0	2	40.0
Gentamicin	A	IV	1	16.7	4	66.7	1	16.7
** *Total* **			23	17.0	57	42.2	55	40.7

NB: A total of 105 patients analyzed as insufficient information was available in the notes from 3 patients; IV = intravenous, O = oral; *** according to WHO AWaRe classification [18], i.e., A: Access, W: Watch, N/A: not available.

**Table 4 antibiotics-14-00709-t004:** Culture and sensitivity testing.

Microbiological Test Characteristics	Variable	(*n*)	(%)
Microbiological test *(n = 108)*	Yes	100	92.6
	No	8	7.4
Time of sampling test *(n = 100)*	Date of entry to hospital After entry hospital day	98 2	98.0 2.0
Tissue for testing *(n = 160)*	Throat Blood Other	97 34 29	60.6 21.6 18.1
Result *(n = 160)*	Positive	47	29.4
	Negative	113	70.6
Results of testing *(n = 47)*	Bacteria	40	85.1
	Virus	7	14.9
Bacterial strains identified *(n = 40)*	*H.influenzae*	25	53.2
	*M.catarrhalis*	12	25.5
	*S.aureus*	2	4.3
	*S.pneumoniae*	1	2.1
	*M.pneumoniae*	0	0.0
	*P.aeruginosa*	0	0.0
Time received microbiological test *(n = 47)*	<3 days	20	42.6
	3 days	16	34.0
	4 days	10	21.3
	5 days	1	2.1

**Table 5 antibiotics-14-00709-t005:** Barriers to prescribing empirical antibiotic therapy adherent to the guidelines.

Barriers to Adherence to Recommended Initial Antibiotic Selection		Comment Transcripts
Internal barriers	Doctors′ experience and prescription habits (5/5)	“The time of diagnosing pneumonia often does not align with the patient′s hospital admission, as many patients have already undergone prior treatment. This makes it challenging to apply standard guidelines accurately, re-sulting in a naturally low compliance rate.” [Dr1]; “Like here is a central hospital, patients have used many antibiotics so of course there are no initial options like simple amoxicillin.” [Dr2].
	Doctor′s treatment perspective: “Recommendations are just recom-mendation”. (2/5)	“Treatment regimens are often theoretical and meant for reference, as there are differences between reality and actual patient populations. Clinical practice requires flexibility, so directly applying theoretical guidelines of-ten leads to low compatibility” [Dr4].
External barriers	Inadequate antibiotics supply (5/5)	“A major challenge is the inconsistent availability of medicines, with shortages and unexpected stockouts forcing patients to switch up to three types of medicine within a week. Often, only limited antibiotic options are available, such as third-generation cephalosporins” [Dr3]. “The only antibiotic covered by health insurance is aug-mentin, but the quantity is very limited. When the patient is hospitalized, there is a group of third-generation ceph-alosporin antibiotics, there is almost no second choice” [Dr5].
	Outdated guidelines (3/5)	“Regular updates to guidelines would greatly assist doc-tors, but in Vietnam, updates are infrequent, with some regimens remaining unchanged for years. Pediatric guidebooks are only published every few years, causing delays in adopting global advancements. Frequent up-dates would improve compliance among doctors” [Dr4].
	Impractical guidelines (2/5)	“In clinical practice, there are discrepancies from the guidelines because patients often have already used multiple medications before hospital admission” [Dr4].
	Inappropriate patient approaches (1/5)	“Classifying patients based on pneumonia severity is typically used in community and lower-level hospitals. In contrast, higher-level hospitals focus more on identifying the underlying causes to choose antibiotics, rather than relying solely on severity, even when the patient exhibits symptoms of respiratory failure from communi-ty-acquired pneumonia” [Dr1].

NB: Numbers refer to the number of pediatricians (out of 5) commenting on these themes.

## Data Availability

Additional data is available on reasonable request from the corresponding authors.

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
