# Peer review of "Barriers to Compliance with National Guidelines Among Children Hospitalized with Community-Acquired Pneumonia in Vietnam and the Implications"

_antibiotics, 2025, doi:10.3390/antibiotics14070709_

Round 1

Reviewer 1 Report

Comments and Suggestions for Authors

This is an interesting study following a mixed methods approach to evaluate the appropriateness of antibiotic use for community-acquired pneumonia among hospitalized children <5 years of age and barriers/facilitators affecting prescription habits in a large pediatric hospital in Vietnam.

Overall the study has meaningful findings that in many cases confirm or augment the findings of previous studies in this country. The narrative parts of the manuscript contain sufficient information and comparisons, providing a clear depiction of the present study within the current landscape in Vietnam.

Despite the study advantages, I have several comments, my main ones being 1.need to add more details in the methodology, as in many cases, the methods are induced by the findings and 2.the need for improving the quality of scientific language across the manuscript, in order to make the study more readable and to better highlight its main findings. In many instances, there is repetition of statements, often repeating the same information in different ways. Some sections are longer than needed due to this repetition, while the content of several statements is unclear. Improvement of quality of English is necessary.

Please also align the content of the abstract with the content of the text.

Specific comments below:

Line 33: avoid starting sentences with numbers.

Line 63, 77: please correct or explain the word “appreciable”.

Lines 40-43: please re-evaluate the conclusion; what’s the relevance between the study aims and findings and the conclusions written in the abstract? Also, the statement about improving antibiotic consumption in the community is irrelevant at this point.

Lines 88-92: please rephrase for clarity. In fact, this and the next paragraph need rephrasing to avoid duplicate and contradictory statements.

Lines 133-164: again, there’s an overflow and duplication of statements around the same issues. Considerable rephrasing is necessary.

Line 202-2023: these 2 percentages add up to more than 100%

Line 205-208: this is unclear. From what I understand, they were discharged either after complete resolution of symptoms, or after significant improvement, correct? Please amend accordingly.

Table 2: can authors briefly describe the most commonly prescribed antibiotics and antibiotic combinations?

Methodology requires further details. In many parts, findings are reported without a corresponding explanation in the methods. Some examples to add in the methods sections: which guidelines were used for assessment of treatment? How was appropriate dosage of antibiotics assessed? What information was sought from the patient files? What questions did the interviews include? How were the interviews performed? What was the response rate of physicians? Please also mention the Aware classification in the methods that was used to classify the antibiotics used.

Discussion section has some large paragraphs that can be broken down.

Line 361: how were these 5 doctors selected?

Lines 407-415: please improve quality of presentation in conclusions. For example: “in this hospital in Vietnam and wider” – what do authors mean by “wider”? Please also adjust conclusions to be similar in the text and in the abstract. Currently, it seems that the abstract focuses on different findings compared to the main text.

Comments on the Quality of English Language

Improvement of quality of English is necessary

Author Response

Comments and Suggestions for Authors

A) General

1) This is an interesting study following a mixed methods approach to evaluate the appropriateness of antibiotic use for community-acquired pneumonia among hospitalized children <5 years of age and barriers/facilitators affecting prescription habits in a large pediatric hospital in Vietnam. Overall the study has meaningful findings that in many cases confirm or augment the findings of previous studies in this country. The narrative parts of the manuscript contain sufficient information and comparisons, providing a clear depiction of the present study within the current landscape in Vietnam.

Author comments: Thank you for your comments – appreciated

2) Despite the study advantages, I have several comments, my main ones being 1.need to add more details in the methodology, as in many cases, the methods are induced by the findings and 2.the need for improving the quality of scientific language across the manuscript, in order to make the study more readable and to better highlight its main findings. In many instances, there is repetition of statements, often repeating the same information in different ways. Some sections are longer than needed due to this repetition, while the content of several statements is unclear. Improvement of quality of English is necessary.

Author comments: Thank you for this. We have now amended the Introduction to focus on key points including key observations/ comments regarding Vietnam incorporating concerns with current prescribing habits, including for CAP, as well as resistance patterns to set the scene for this paper – as well as possible ways forward including ASPs (which have worked well in this area for other LMICs, etc.). We have also refined the Discussion to increase its focus as well as add considerably more details to the Methodology section. trust this is now acceptable. We have also improved the English with the help of one of the co-authors who is a native English speaker with over 500 publications in peer-reviewed Journals in recent years. We trust this is now acceptable.

3) Please also align the content of the abstract with the content of the text.

Author comments: Thank you for pointing this out. We have now addressed this, and hope this is now OK.

B) Specific comments below:

1) Line 33: avoid starting sentences with numbers.

Author comments: Thank you – now addressed

2) Line 63, 77: please correct or explain the word “appreciable”.

Author comments: Thank you – now addressed

3) Lines 40-43: please re-evaluate the conclusion; what’s the relevance between the study aims and findings and the conclusions written in the abstract? Also, the statement about improving antibiotic consumption in the community is irrelevant at this point.

Author comments: Thank you. We would like to keep this point because if trainee ambulatory care physicians get into bad habits regarding the prescribing of antibiotics during their training in hospitals – this will spill over into their practice once qualified and treating children in the community. Conversely if they are appropriately trained in antibiotic use/ misuse whilst training in teaching hospitals – hopefully these practices will remain with them once qualified (as discussed in original lines 162-164). We hope that you now agree with us in the expanded abstract.

4) Lines 88-92: please rephrase for clarity. In fact, this and the next paragraph need rephrasing to avoid duplicate and contradictory statements.

Author comments: Thank you – now addressed.

5) Lines 133-164: again, there’s an overflow and duplication of statements around the same issues. Considerable rephrasing is necessary.

Author comments: Thank you – now hopefully addressed.

6) Line 202-203: these 2 percentages add up to more than 100%

Author comments: Thank you for pointing this out. This has now been corrected.

7) Line 205-208: this is unclear. From what I understand, they were discharged either after complete resolution of symptoms, or after significant improvement, correct? Please amend accordingly.

Author comments: Thank you – we have now edited this to reduce possible confusion, and hope this is now OK.

8) Table 2: can authors briefly describe the most commonly prescribed antibiotics and antibiotic combinations?

Author comments: Thank you for your comments – now added in.

9) Methodology requires further details. In many parts, findings are reported without a corresponding explanation in the methods. Some examples to add in the methods sections: which guidelines were used for assessment of treatment? How was appropriate dosage of antibiotics assessed? What information was sought from the patient files? What questions did the interviews include? How were the interviews performed? What was the response rate of physicians? Please also mention the Aware classification in the methods that was used to classify the antibiotics used.

Author comments: Thank you for pointing this out. We have now included considerably more information covering these points in the updated Methodology section, and trust this is now acceptable.

10) Discussion section has some large paragraphs that can be broken down.

Author comments: Thank you – now hopefully addressed.

11) Line 361: how were these 5 doctors selected?

Author comments: Thank you for pointing this out. We have now clarified this in the Methodology section, and hope this is now OK.

12) Lines 407-415: please improve quality of presentation in conclusions. For example: “in this hospital in Vietnam and wider” – what do authors mean by “wider”? Please also adjust conclusions to be similar in the text and in the abstract. Currently, it seems that the abstract focuses on different findings compared to the main text.

Author comments: Thank you. We have now updated the Conclusion to improve the focus and suggested activities. We hope this is now acceptable.

13) Comments on the Quality of English Language: Improvement of quality of English is necessary

Author comments: Thank you. As previously stated, the paper has now been updated with the help of one of the co-authors who is a native English speaker with over 500 publications in peer-reviewed Journals in recent years. We trust this is now OK.

Reviewer 2 Report

Comments and Suggestions for Authors

1. The introduction provides relevant background but is too lengthy and repeats concepts, especially around AWaRe classification and CAP prevalence. I suggest to little bit rephrase and give only brief explanation like in lines 42–47 and 53–59. 

2. Inclusion criteria are presented, but the rationale for the 3-day hospital stay is not explained. Justify why this threshold was selected—was it related to stable data collection or treatment duration?

3. Line 95-100, the term "compliance" must be described clearly, what is compliance, is it about correct choice, dose, route, and duration?

4. The interview component, though insightful, is based on only five physicians without justification of thematic saturation. Thematic analysis methodology should be described more explicitly, and the findings would benefit from richer, more diverse quote representation.

5. I suggest to conduct statistical analysis between severe and non-severe pneumonia, or between compliant and non-compliant cases

6. The discussion provides a good summary of the findings, but it often restates the results rather than more depth discussion on their deeper implications. Suggestion to give more insight or discussion on how physician behaviors are shaped by systemic constraints—like limited training opportunities, resource shortages, or lack of updated clinical guidance.

7. Give summarize 2–3 action points in conclusion section, e.g., update national guidelines every 5 years, develop dosing training for pediatricians. 

Author Response

Comments and Suggestions for Authors

1. The introduction provides relevant background but is too lengthy and repeats concepts, especially around AWaRe classification and CAP prevalence. I suggest to little bit rephrase and give only brief explanation like in lines 42–47 and 53–59. 

Author comments: Thank you for this. We have now cut down on the Introduction. However, retained a number of observations/ comments regarding Vietnam including concerns with current prescribing habits, including for CAP, as well as resistance patterns to set the scene for this paper – as well as possible ways forward including ASPs (which have worked well in this area for other LMICs, etc.). We trust this is now acceptable.

2. Inclusion criteria are presented, but the rationale for the 3-day hospital stay is not explained. Justify why this threshold was selected—was it related to stable data collection or treatment duration?

Author comments: Thank you for pointing this out. We have now included an explanation for this when mentioning this in the Methodology section, and hope this is now OK.

3. Line 95-100, the term "compliance" must be described clearly, what is compliance, is it about correct choice, dose, route, and duration?

Author comments: Thank you – now inserted into the Methodology.

4. The interview component, though insightful, is based on only five physicians without justification of thematic saturation. Thematic analysis methodology should be described more explicitly, and the findings would benefit from richer, more diverse quote representation.

Author comments: Thank you – now updated in the Methodology. We hope this is now acceptable.

5. I suggest to conduct statistical analysis between severe and non-severe pneumonia, or between compliant and non-compliant cases

Author comments: Thank you - In Table 2, the data were analyzed with the antibiotic regimens separately analysed according to the severity of CAP. We have though consolidated the findings especially in the in the Discussion in view of the similarities between the two groups – with the main message to urgently address given considerable concerns with antibiotic utilisation patterns in Vietnam/ coupled with high AMR rates is one of current low compliance with guidelines, high use of IV administration and considerable prescribing of Watch antibiotics – all of which need to be addressed going forward. We trust this is acceptable to you.

6. The discussion provides a good summary of the findings, but it often restates the results rather than more depth discussion on their deeper implications. Suggestion to give more insight or discussion on how physician behaviors are shaped by systemic constraints—like limited training opportunities, resource shortages, or lack of updated clinical guidance.

Author comments: Thank you – we have now consolidated key parts of the Discussion as well as suggested ways forward for all key stakeholder groups including improved education of physicians in University and carried forward in CPD activities. We hope this is now OK.

7. Give summarize 2–3 action points in conclusion section, e.g., update national guidelines every 5 years, develop dosing training for pediatricians. 

Author comments: Thank you – the Conclusion has now been updated (with input from also from other Reviewers). We trust this is now acceptable.

Reviewer 3 Report

Comments and Suggestions for Authors

This is a mixed-methods study in a major hospital assessed CAP management and guideline adherence. Among 108 pediatric patients (mean age 27.94 months), 82.4% had non-severe CAP, yet all received antibiotics—91.7% 3rd-generation cephalosporins, primarily cefoperazone/sulbactam (48.2%). Most (73.3%) antibiotics were WHO ‘Watch’-listed, 96.1% exceeded recommended doses, and only 4.63% followed guidelines. Interviews revealed barriers to adherence, highlighting inappropriate antibiotic use.

The authors have conducted a commendable study that highlights the urgent need for antimicrobial stewardship programs to align treatment practices with established guidelines, curb antimicrobial resistance (AMR), and improve antibiotic use within communities. However, I recommend that the Introduction section be shortened and made more concise to enhance clarity and focus. Additionally, the Conclusion section should be revised to include more forward-looking statements, emphasizing potential future research directions and practical implications.

Author Response

Comments and Suggestions for Authors

1) This is a mixed-methods study in a major hospital assessed CAP management and guideline adherence. Among 108 pediatric patients (mean age 27.94 months), 82.4% had non-severe CAP, yet all received antibiotics—91.7% 3rd-generation cephalosporins, primarily cefoperazone/sulbactam (48.2%). Most (73.3%) antibiotics were WHO ‘Watch’-listed, 96.1% exceeded recommended doses, and only 4.63% followed guidelines. Interviews revealed barriers to adherence, highlighting inappropriate antibiotic use.

Author comments: Thank you for this – appreciated!

2) The authors have conducted a commendable study that highlights the urgent need for antimicrobial stewardship programs to align treatment practices with established guidelines, curb antimicrobial resistance (AMR), and improve antibiotic use within communities.

Author comments: Thank you for this – appreciated!

3) However, I recommend that the Introduction section be shortened and made more concise to enhance clarity and focus. Additionally, the Conclusion section should be revised to include more forward-looking statements, emphasizing potential future research directions and practical implications.

Author comments: Thank you for this. As seen, we have now consolidated the Introduction to focus especially on key points for Vietnam including current high AMR rates, concerns with antibiotic prescribing including for patients with CAP and possible ways forward including ASPs. The Conclusion section has also been updated, and we hope both are now acceptable.

Round 2

Reviewer 1 Report

Comments and Suggestions for Authors

Authors have addressed several comments, however I’m afraid that there are still points of uncertainty.

For example, the word “appreciable” is still repeated across the manuscript, but its meaning remains unclear.

One paragraph that I had commented on previously, still is unclear. Specifically, line 189 “Children were prescribed antibiotics on average for 6.00 (5.25 – 7.00) days, and typically discharged on average after 7.00 (6.00 – 8.00) days either because their symptoms had disappeared; alternatively with sufficient improvement.” Is the average duration 6 or 7 days?

Brief description of table 2 not added, despite authors replying that they have.

Lines 417-418: there’s no evidence that the findings in this hospital reflect the situation in other hospitals; in contrast, a common limitation of a single-center study is that it lacks generalizability.

Line 422: why would the authors recommend refining national guidelines every 5 years?

Conclusion paragraph is poor and generic. I’d suggest to keep it short and simple and just relate to the main study findings.

Comments on the Quality of English Language

Improvements still needed throughout.

Author Response

Comments and Suggestions for Authors

1) Authors have addressed several comments, however I’m afraid that there are still points of uncertainty.

Author comments: Thank you for this. Hopefully, we can satisfactorily address your remaining comments.

2) For example, the word “appreciable” is still repeated across the manuscript, but its meaning remains unclear.

Author comments: Thank you – we have updated this accordingly.

3) One paragraph that I had commented on previously, still is unclear. Specifically, line 189 “Children were prescribed antibiotics on average for 6.00 (5.25 – 7.00) days, and typically discharged on average after 7.00 (6.00 – 8.00) days either because their symptoms had disappeared; alternatively with sufficient improvement.” Is the average duration 6 or 7 days?

Author comments: Thank you – now clarified. Hopefully, now OK.

4) Brief description of table 2 not added, despite authors replying that they have.

Author comments: Apologies for this – hopefully now addressed.

5) Lines 417-418: there’s no evidence that the findings in this hospital reflect the situation in other hospitals; in contrast, a common limitation of a single-center study is that it lacks generalizability.

Author comments: Thank you for this. We have now added to the limitations by saying that the findings from this study build on other similar studies in Vietnam. In addition – in view of the characteristics of this hospital – if there are identified problems in this hospital – these will be replicated throughout Vietnam. We trust this is now acceptable.

6) Line 422: why would the authors recommend refining national guidelines every 5 years?

Author comments: Thank you – now updated

7) Conclusion paragraph is poor and generic. I’d suggest to keep it short and simple and just relate to the main study findings.

Author comments: Thank you for pointing this out – now substantially added to. We hope this is now OK.

8) Comments on the Quality of English Language - Improvements still needed throughout.

Author comments: Thank you – now hopefully addressed.

Reviewer 2 Report

Comments and Suggestions for Authors

Thank you for the revision, I think all concern have been answered and impelemented properly in the manuscript

Author Response

Comments and Suggestions for Authors

Thank you for the revision, I think all concern have been answered and implemented properly in the manuscript

Author comments: Thank you – appreciated!
